# *Pseudorabies Virus* IE180 Inhibits Virus Replication by Activating the Type I Interferon Pathway

**DOI:** 10.3390/microorganisms13061397

**Published:** 2025-06-16

**Authors:** Feiyang Zheng, Jingjing Song, Xuan Chen, Dongyue Xing, Rulan Bai, Changyong Cheng, Jin Yuan, Rui Zhang

**Affiliations:** 1National Key Laboratory of Veterinary Public Health and Safety, College of Veterinary Medicine, China Agricultural University, Beijing 100193, China; 15029490340@163.com (F.Z.); sjjndu@163.com (J.S.); cx18810529379@163.com (X.C.); 18691950047@163.com (D.X.); rulan@cau.edu.cn (R.B.); 2College of Veterinary Medicine, China Agricultural University, Beijing 100193, China; 3Key Laboratory of Applied Biotechnology on Animal Science & Veterinary Medicine of Zhejiang Province, Zhejiang A&F University, 666 Wusu Street, Lin’an District, Hangzhou 311300, China; lamge@zafu.edu.cn; 4Zhejiang Engineering Research Center for Veterinary Diagnostics & Advanced Technology, Zhejiang A&F University, 666 Wusu Street, Lin’an District, Hangzhou 311300, China; 5College of Veterinary Medicine, Zhejiang A&F University, 666 Wusu Street, Lin’an District, Hangzhou 311300, China; 6College of Animal Science and Veterinary Medicine, Henan Agricultural University, Zhengzhou 450046, China; yuanjin@henau.edu.cn

**Keywords:** *pseudorabies virus*, IE180, type I interferon pathway, antiviral immunity

## Abstract

The immediate-early protein IE180 of *pseudorabies virus* (PRV) is a multifunctional regulator of viral and host gene expression. However, its role in modulating antiviral immune responses remains poorly understood. Here, we demonstrate that IE180 overexpression significantly inhibits PRV and *H1N1 influenza virus* replication in Hep2 and A549 cells, respectively. Mechanistically, IE180 activates the type I interferon (IFN-I) pathway by enhancing IFN-β promoter activity and IFN transcription, leading to upregulated expression of interferon-stimulated genes (*ISGs*). Notably, IE180 failed to suppress PRV or H1N1 replication in Vero cells, which lack functional IFN-I signaling, confirming the dependence of its antiviral function on the IFN-I pathway. Domain mapping revealed that the ICP4-Like2 domain of IE180 is critical for IFN-β activation and antiviral activity. These findings establish IE180 as a novel viral immunomodulator that activates host innate immunity to restrict viral replication, providing insights into PRV-host interactions and potential therapeutic strategies.

## 1. Introduction

*Pseudorabies virus* (PRV), a member of the *Alphaherpesvirinae* subfamily, is the causative agent of pseudorabies (PR), an acute infectious disease that primarily affects pigs and poses significant threats to the swine industry. PRV can also infect a wide range of other mammals, including ruminants, carnivores, and rodents, often resulting in fatal neurological disorders in these non-natural hosts [1,2]. The PRV genome encodes more than 70 proteins, among which immediate early protein IE180 is the sole immediate early gene product, which is a functional homolog of herpes simplex virus ICP4 [3]. IE180 functions as a key transcriptional regulator for initiating PRV replication by transactivating early and late genes [4,5]. Notably, a truncated IE180 mutant (dIN454-C1081), lacking the acidic transcriptional activation domain, acts as a dominant-negative repressor by suppressing viral IE gene transcription [6]. Additionally, IE180 exhibits cross-regulatory activity with other alphaherpesvirus homologs, such as negatively regulating the transcription activation of herpes simplex virus type 1 (HSV-1) ICP4 promoter [7]. These findings highlight IE180’s dual regulatory role, yet the molecular mechanisms underlying its context-dependent functions remain unclear.

Beyond its well-established role in viral transcription, IE180 interacts with multiple host pathways, influencing both viral replication and cellular processes. For example, IE180 inhibits the phosphorylation of eukaryotic translation initiation factor eIF2α by recruiting the cellular phosphatase PP1, thereby sustaining viral protein synthesis under stress conditions [8]. Additionally, when expressed under tumor-specific promoters such as carcinoembryonic antigen (CEA), IE180 overexpression induces apoptosis in cancer cells [9]. A newly identified mechanism reveals that IE180 hijacks host G3BP1/2 proteins into the nucleus to suppress stress granule (SG) formation during PRV infection, thereby facilitating viral replication [10]. Despite these findings, the immunomodulatory functions of IE180 remain largely unexplored.

The type I interferon (IFN-I) pathway serves as a cornerstone of the host defense against viral infections [11]. Upon detection of viral components, host cells produce IFN-I, which initiates the phosphorylation and subsequent activation of signal transducer and activator of transcription (STAT) proteins. This signaling cascade ultimately drives the assembly of the Interferon Stimulated Gene Factor 3 (ISGF3) complex, composed of phosphorylated STAT1 (P-STAT1), phosphorylated STAT2 (P-STAT2), and interferon regulatory factor 9 (IRF9). The ISGF3 complex then translocates to the nucleus to induce the expression of interferon-stimulated genes (ISGs), which encode potent antiviral effectors [12,13,14]. While many viral proteins antagonize IFN-I signaling to facilitate infection, certain viral factors paradoxically enhance this pathway [12,15].

In this study, we investigate the role of PRV IE180 in modulating IFN-I responses and its functional impact on viral replication. We found that IE180 overexpression inhibited the replication of both PRV and an unrelated virus, H1N1 influenza, in IFN-competent cells by promoting IFN-β promoter activity and enhancing the expression of multiple ISGs. Importantly, this antiviral effect was absent in Vero cells lacking IFN-I signaling, confirming its dependence on intact host IFN signaling. Domain-mapping experiments further identified the ICP4-Like2 region of IE180 as essential for its ability to stimulate IFN-β and exert antiviral effects. These findings position IE180 as a bifunctional regulator at the virus-host interface, offering novel insights into PRV pathogenesis and highlighting potential targets for antiviral strategies.

## 2. Materials and Methods

### 2.1. Cell Culture and Viruses

HEK293T cells (human embryonic kidney, ATCC #CRL-3216, Manassas, VA, USA), PK15 cells (porcine kidney cells, ATCC #CCL-33, Manassas, VA, USA), Hep2 cells (human epithelial type 2, and a kind gift from Xiaojia Wang, were cultured in Dulbecco’s modified Eagle’s medium (DMEM, Invitrogen, Waltham, MA, USA). MDCK cells (Madin-Darby Canine Kidney, ATCC #CRL-2935, Manassas, VA, USA) were cultured in a minimum essential medium (MEM, Invitrogen, Waltham, MA, USA). All the culture mediums contain 10% fetal bovine serum (FBS, Gibco, Thermo Fisher Scientific, Waltham, MA, USA) and 1% penicillin/streptomycin (Gibco, Thermo Fisher Scientific). All cells were maintained under 5% CO_2_ in a constant-temperature incubator at 37 °C.

The PRV BarthaK61 strain vaccine (lot number 2012002) was purchased from Weike Biotech Co., Harbin, China. Influenza A virus (H1N1) strain A/Puerto Rico/8/34 (PR8) was obtained from Juan Pu (China Agriculture University, Beijing, China).

### 2.2. Antibodies, Reagents, and Plasmids

Antibodies used in this study were p-STAT1 (9167S), p-STAT2 (88410), and HA (C29F4) from Cell Signaling Technology (Danvers, MA, USA); Flag (F1804, Sigma-Aldrich, St. Louis, MO, USA), ISG15 (sc-166755, Santa Cruz Biotechnology, Dallas, TX, USA), tubulin (PM054, MBL, Woburn, MA, USA), GAPDH (ab8245, Abcam, Cambridge, UK), and influenza A virus NP (A01506-40, GenScript, Piscataway, NJ, USA). The antibodies against viral proteins IE180, VP5, and US3 were raised in mice following a standard procedure as described previously [16]. HRP-conjugated goat anti-mouse and anti-rabbit secondary antibodies were from Santa Cruz Biotechnology (Dallas, TX, USA). The full-length and truncated forms of IE180 were amplified from the BarthaK61 genome and cloned into the pRK5 vector with an N-terminal Flag tag. Recombinant human IFN-α was purchased from Novoprotein (Shanghai, China).

### 2.3. Luciferase Reporter Assay

The 293T cells were seeded in 24-well plates and transfected with an EV or plasmids expressing viral proteins together with 100 ng of luciferase reporter plasmids ISRE (pGL3-ISRE-Luc) (Firefly luciferase, Promega, Madison, WI, USA) or IFN-β (pGL3-IFN-β-Luc), or MBT1 (pGL3-MBT1-Luc) and 10 ng of pRL-TK (Renilla luciferase, Promega, Madison, WI, USA) by using jetPRIME DNA transfection reagent (Polyplus-transfection, Illkirch, Grand Est, France). The total amount of DNA was made constant by adding the pRK5 vector. Then, 24 h after transfection, cells were incubated in the media containing human IFN-α (1000 U/mL) for 12 h and were harvested and analyzed for luciferase activities using a Dual-Luciferase reporter assay kit (Promega, Madison, WI, USA), according to the manufacturer’s instructions.

### 2.4. Quantitative Real-Time PCR, qPCR

Total RNA was extracted using an RNA extraction kit (Magen Biotech, Guangzhou, China) following the manufacturer’s protocol. For each sample, 0.8 μg of RNA was reverse-transcribed using Moloney murine leukemia virus reverse transcriptase (M-MLV RT, Promega, Madison, WI, USA) and an oligo(dT)18 primer (Takara, Kusatsu, Shiga, Japan). qPCR was performed using the UltraSYBR mixture (CW2601H, Beijing CoWin Biotech, Beijing, China) and the ViiATM7 RT-PCR system (Applied Biosystems, Thermo Fisher Scientific, Waltham, Ma, Usa). Relative gene expression levels were calculated using the ΔΔCt (2^−ΔΔCt^)method. GAPDH was used as a housekeeping gene to normalize the relative mRNA expressions of target genes. Each reaction was performed in triplicate, and the mean Ct value was used for analysis.

Gene-specific primers for qPCR were designed using Primer-BLAST (NCBI) to ensure specificity and optimal melting temperatures. The primers were synthesized by Sangon Biotech, China, and their sequences are listed as follows: IFN-β (5′-AGCCAAAAGGGTCATCATCTC-3′; 5′-GGACTGTGGTCATGAGTCCTTC-3′), IFN-α (5′-AATGACAGAATTCATGAAAGCGT-3′; 5′-GGAGGTTGTCAGAGCAGA-3′), IRF9 (5′-GCCCTACAAGGTGTATCAGTTG-3′; 5′-TGCTGTCGCTTTGATGGTACT-3′), ISG15 (5′-CAGATCACCCAGAAGATCG-3′; 5′-CCCTTGTTATTCCTCACCAG-3′), ISG56 (5′-ACACCTGAAAGGCCAGAATGAGGA-3′; 5′-TGCCAGTCTGCCCATGTGGTAATA-3′), GAPDH (5′-AGCCAAAAGGGTCATCATCTC-3′; 5′-GGACTGTGGTCATGAGTCCTTC-3′).

### 2.5. Western Blot Analysis

Whole-cell lysates were prepared in lysis buffer (50 mM Tris-Cl at pH 8.0, 150 mM NaCl, 1.0% Triton X-100, 10% glycerol, 20 mM NaF, 1 mM DTT, and 1× complete protease mixture, Roche, Basel, Switzerland) supplemented with protease inhibitors. The proteins were resolved by 10% sodium dodecyl sulfate-polyacrylamide gel electrophoresis (SDS-PAGE) and transferred to nitrocellulose membranes, which were blocked with 5% skim milk in PBST (PBS containing 0.5% Tween 20) for 2 h at room temperature and were then incubated with specific primary antibodies overnight at 4 °C, followed by secondary antibodies for 45 min at room temperature. The reactive protein bands were visualized using an enhanced chemiluminescence (ECL) reagent with a Tanon-5200 luminescent imaging workstation (Tanon Science & Technology Co., Ltd., Shanghai, China).

### 2.6. Virus Infection and Titer Determination

The PK15 cells were infected with PRV for 1 h, washed with phosphate-buffered saline (PBS), and incubated in complete DMEM supplemented with 5% FBS for the indicated durations. The supernatants were collected for titer determination, and the remaining cells were used for Western blot analysis. The viral yield was determined by tittering in PK15 cells. Briefly, the collected supernatants from virus-infected PK15 cells were cleared of cell debris by centrifugation and then used to infect PK15 cells in 10-fold serial dilutions in a 96-well plate. Then, 72 to 96 h post-infection, the cytopathic effect (CPE) in each well was observed and scored. The 50% tissue culture infective dose (TCID50)/mL was calculated by the Reed–Muench method.

For H1N1 infection, MDCK cells were washed with PBS and infected with the PR8 strain at a multiplicity of infection (MOI) of 1. After 1 h of adsorption at 37 °C, the inoculum was removed, and cells were maintained in MEM containing 1 μg/mL TPCK-treated trypsin. Supernatants were collected at the indicated time points, and viral titers were determined by plaque assay on MDCK cells. Briefly, serial tenfold dilutions of supernatants were added to confluent MDCK monolayers and incubated for 1 h at 37 °C. The cells were then overlaid with 1.5% agarose in MEM containing 1 μg/mL TPCK-treated trypsin. After 72 h, plaques were visualized by crystal violet staining and counted to calculate plaque-forming units (PFU) per milliliter.

### 2.7. Chromatin Immunoprecipitation (ChIP) Assay

The ChIP assay was performed using a Magna ChIP A/G kit (17-10085, EMD Millipore, Burlington, MA, USA) following the manufacturer’s instructions. Briefly, HEK293T cells were fixed with 1% paraformaldehyde and lysed with SDS lysis buffer, followed by sonication and immunoprecipitation with anti-Flag, anti-IE180, or control IgG Abs. Then, the protein/DNA complexes were reverse cross-linked to free DNA, and the specific DNA fragments were analyzed by qRT-PCR and normalized to input from the same cells. Specific primers used for qRT-PCR include IFN-β (5′-ATTCCTCTGAGGCAGAAAGGACCA-3′; 5′-GCAAGATGAGGCAAAGGCTGTCAA-3′) and ISRE (from ISG15, 5′-CGCCACTTTTGCTTTTCCCT-3′; 5′-ATAAGCCTGAGGCACACACG-3′).

### 2.8. Statistical Analysis

Statistical analysis was performed using GraphPad Prism software (version 9.5.1, GraphPad Software, San Diego, CA, USA) to perform analysis of Student’s *t*-test or variance (ANOVA) on at least three independent replicates. *p* values of 0.05 were considered statistically significant for each test. *, *p* < 0.05, **, *p* < 0.01; ***, *p* < 0.001, ****, *p* < 0.0001.

## 3. Results

### 3.1. IE180 Suppresses PRV and H1N1 Replication

Previous studies have shown that a truncated IE180 mutant (dIN454-C1081) represses PRV replication by inhibiting viral gene transcription [6], while IE180 also cross-regulates HSV-1 ICP4 [7], suggesting a potential antiviral role. To further investigate IE180’s antiviral activity, Hep2 and A549 cells were transfected with empty vector (EV) or IE180-expressing plasmids, followed by infection with PRV (MOI = 1) or influenza A virus H1N1 (PR8 strain, MOI = 1). Western blot analysis revealed that IE180 significantly reduced the expression of PRV protein (US3) in Hep2 cells at 12 and 24 h post-infection (h.p.i.) compared to EV-transfected controls (Figure 1A). Consistently, viral titers in supernatants were markedly decreased in IE180-expressing cells (Figure 1B). Similarly, the overexpression of IE180 in A549 cells suppressed H1N1 replication, as evidenced by reduced viral nucleoprotein (NP) levels (Figure 1C) and lower viral titers (Figure 1D). These results demonstrate that the ectopic IE180 expression inhibits both homologous (PRV) and heterologous (H1N1) virus replication, highlighting its antiviral potential.

### 3.2. IE180 Activates IFN-I Signaling Pathways

Given the antiviral activity of IE180 against both DNA (PRV) and RNA (H1N1) viruses, we hypothesized that IE180 may enhance host innate immune defenses. To test this, we evaluated its impact on IFN-I pathway activation. HEK293T cells were cotransfected with IE180 and an IFN-stimulated response element (ISRE)-driven luciferase reporter (pGL3-ISRE-Luc). After 24 h, the cells were treated with IFN-α for 12 h, followed by luciferase activity analysis. As shown in Figure 2A, IE180 robustly activated ISRE compared to EV controls, regardless of IFN treatment. To determine whether IE180 directly induces IFN-β production, we tested its effect on an IFN-β promoter-driven luciferase reporter. IE180 significantly activated the IFN-β promoter compared to the control vector (Figure 2B, left panel), whereas it had no effect on the HIF1α-induced MBT1 (histone methyl-lysine binding protein 3) promoter (Figure 2B, right panel), confirming that IE180 specifically activates IFN-I transcription. In Hep2 cells, IE180 overexpression dramatically upregulated IFN-β mRNA levels, along with IFN-stimulated genes (IRF9, ISG15, and ISG56), while having only a minimal effect on IFN-α mRNA (Figure 2C), suggesting that it may specifically target IFN-β. Western blot analysis further showed that IE180 enhanced STAT1 and STAT2 phosphorylation, as well as IRF9 and ISG15 expression, mimicking the effects of the dsDNA mimic Poly(dA:dT) and IFN-α treatment (Figure 2D). Time-course experiments revealed a progressive increase in STAT1/2 phosphorylation over 24 h in Hep-2 cells expressing IE180 (Figure 2E), and a similar trend was observed in MDCK cells (Figure 2F), indicating that IE180-induced type I interferon signaling activation is largely cell type-independent. Interestingly, the expression level of IE180 under the EV (negative control) condition appeared higher in MDCK cells than in Hep-2 cells. This discrepancy may be due to differences in viral infection efficiency or distinct kinetics of viral protein expression in infected cells. These observations suggest that while the downstream signaling response to IE180 is comparable, the dynamics of IE180 expression may vary across different host cell types.

### 3.3. IE180’s Antiviral Function Requires Intact IFN-I Signaling

To determine whether IE180’s antiviral effects rely on IFN-I signaling, we examined its activity in Vero cells, which lack a functional IFN-I response. In contrast to IFN-competent Hep2 and A549 cells, IE180 failed to suppress PRV or H1N1 replication in Vero cells, with viral protein levels (US3, VP5, and NP) and titers remaining indistinguishable from EV controls (Figure 3A,B). This stark contrast confirms that IE180’s antiviral activity is IFN-I dependent.

### 3.4. The ICP4-Like2 Domain of IE180 Mediates IFN-β Activation and Antiviral Activity

To determine the structural basis of IE180’s immunostimulatory function, we generated a series of truncation mutants based on its functional domains (Figure 4A) and measured their ability to activate IFN-signaling in the IFN-β and ISRE reporter assays. Full-length IE180 and its ICP4-Like2 domain (residues 930–1081) activated the IFN-β promoter by 42.9-fold and 9.8-fold, respectively, whereas mutants lacking this domain showed no significant activity (Figure 4B, left panel). A similar trend was observed for ISRE activation (Figure 4B, right panel). Chromatin immunoprecipitation (ChIP) assays confirmed that IE180 and ICP4-Like2 bind directly to the IFN-β promoter but not to ISRE (Figure 4C), indicating that the ICP4-Like2 domain plays a key role in IFN-β induction. Functionally, ICP4-Like2 expression alone mirrored the antiviral effects of full-length IE180, reducing both viral protein levels and titers of PRV (Figure 4D) and H1N1 (Figure 4E) by more than 20%. These findings establish ICP4-Like2 as a critical mediator of IFN-β activation and antiviral activity.

## 4. Discussion

Our study identifies IE180 as a unique viral protein with dual roles in viral and host regulation. While IE180 is essential for PRV gene activation, its ability to induce IFN-β transcription contrasts with canonical viral IFN antagonists. This finding aligns with growing evidence that certain viral proteins can activate innate immune pathways, either as an unintended consequence or as a strategic mechanism to regulate viral replication and immune evasion. Similar dual functionalities have been observed in other viruses. For instance, while the NS1 protein of influenza A virus primarily functions as an IFN antagonist, specific mutant forms have been shown to stimulate RIG-I signaling, leading to enhanced IFN-I responses [17]. Similarly, hepatitis C virus (HCV) NS5B activates IFN-β through TLR3/TRIF signaling; however, this activation is counterbalanced by the suppressive effects of NS4A, NS4B, and NS5A, enabling the virus to establish persistent infection [18]. In another case, HIV-1 Vpr induces the expression of interferon-stimulated genes and promotes STAT1 phosphorylation in macrophages, yet paradoxically facilitates the maintenance of viral reservoirs by modulating immune activation [19]. These examples underscore the complex interplay between viral proteins and host immunity, highlighting their context-dependent roles in infection and persistence.

Interestingly, a recent study demonstrated that IE180 interacts with G3BP1/2 proteins to inhibit Stress Granules (SG) formation by sequestering them in the nucleus, which contrasts with its IFN-I-activating role reported here [10]. While our findings establish IE180 as an activator of IFN-I-dependent antiviral responses, the suppression of SGs by IE180 may represent a counterbalancing mechanism to optimize viral fitness. SGs are antiviral hubs that restrict viral translation, and their inhibition by IE180 could mitigate the host’s translational shutdown, thereby supporting viral replication despite IFN-I activation. This duality suggests that IE180 temporally fine-tunes host responses: early IFN-I induction may delay immune overactivation, while late SG suppression facilitates viral proliferation.

The structural basis of these divergent functions warrants further investigation. While the ICP4-Like2 domain is essential for IFN-β activation in our study, the ICP4-Like1 domain mediates G3BP1/2 binding in the SG suppression mechanism [10]. This domain-specific specialization implies that IE180 employs distinct regions to manipulate different host pathways. Future studies should explore whether these activities are mutually exclusive or synergistically regulated during infection, particularly in the context of viral latency-reactivation cycles.

Functionally, ectopic expression of IE180 activated the IFN-β promoter, upregulated IFN transcription, and induced ISG expression, confirming its role as a positive regulator of IFN-I signaling. This activation was biologically significant, as IE180 restricted PRV and H1N1 replication in IFN-competent cells but lost this antiviral effect in Vero cells, which lack a functional IFN response. These findings reinforce the notion that IE180’s antiviral activity is mediated through IFN-I signaling. The ability of IE180 to function both as a transcriptional activator and an immune modulator suggests an evolutionary adaptation of PRV to fine-tune IFN-I responses during early infection. Given that IFN-I signaling not only restricts viral replication but also shapes adaptive immunity, IE180’s activation of this pathway may represent a finely tuned regulatory mechanism. It is plausible that transient IFN-I induction by abundant IE180 expression in the early phase of PRV infection optimizes viral spread by mitigating excessive cytotoxic immune responses while still permitting sufficient viral replication. A similar strategy has been observed in other viral proteins, such as the Ebola virus VP35, which can function as both an IFN antagonist and an activator under different conditions [20,21,22,23,24]. However, an important consideration in our study is the reliance on overexpression systems, raising questions about whether IE180’s IFN-inducing activity is physiologically relevant during natural infection. Although IE180 is highly expressed in the immediate-early phase of PRV infection, its precise role throughout the viral replication cycle remains to be elucidated.

In summary, this study redefines IE180 as a multifunctional regulator of PRV-host interactions, linking viral transcriptional regulation to innate immune modulation. The discovery that IE180 activates IFN-I signaling to restrict viral replication provides novel insights into alphaherpesvirus immunobiology and suggests potential antiviral strategies. Future studies exploring IE180’s interactions with IFN-I pathways, as well as its role in PRV latency and reactivation, will enhance our understanding of PRV pathogenesis and may inform the development of targeted antiviral interventions.

## Figures and Tables

**Figure 1 microorganisms-13-01397-f001:**
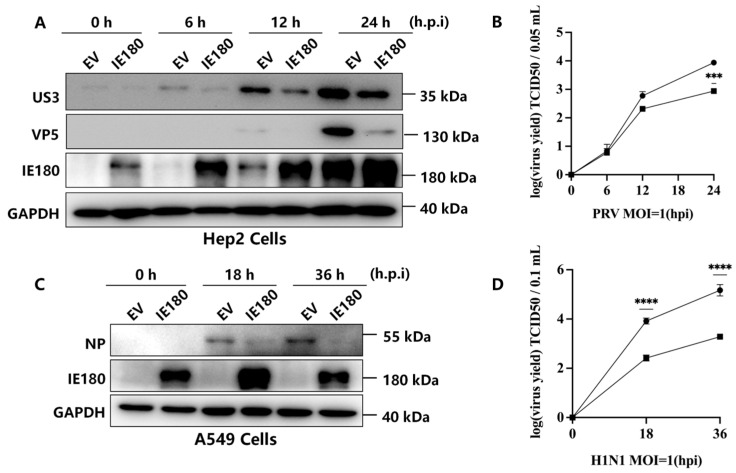
IE180 Overexpression inhibits PRV and H1N1 replication. (**A**,**B**) Hep2 cells were transfected with either an empty vector (EV) or an IE180-expressing plasmid for 24 h, followed by PRV infection (MOI = 1) at the indicated time points. Viral protein levels (VP5, US3) were analyzed by Western blot (**A**), and viral titers in supernatants were measured using the TCID50 assay (**B**). (**C**,**D**) A549 cells were transfected with EV or IE180 plasmids for 24 h, followed by H1N1 (PR8) infection (MOI = 1). Viral nucleoprotein (NP) expression was assessed by Western blot (**C**), and viral titers were quantified by TCID50 (**D**). Data are presented as mean ± SD from three independent experiments. Statistical significance was determined using ANOVA (*** *p* < 0.001, ****, *p* < 0.0001).

**Figure 2 microorganisms-13-01397-f002:**
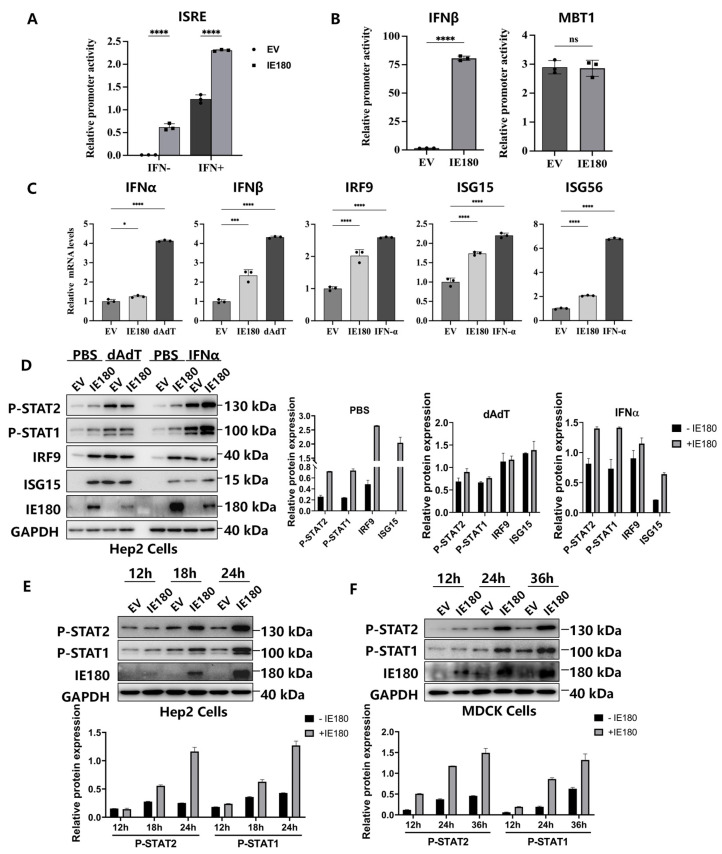
IE180 activates IFN-I signaling. (**A**) IE180 enhances ISRE activation, independent of IFN stimulation. 293T cells were cotransfected with EV or IE180, along with a firefly luciferase reporter driven by ISRE (pGL3-ISRE-Luc) and a constitutively expressed Renilla luciferase plasmid (pRL-TK) as an internal control. After 24 h, cells were treated with IFN-α (1000 U/mL) for 12 h and analyzed for luciferase activity. (**B**) IE180 promotes IFN-β activation. 293T cells were cotransfected with EV or IE180, together with luciferase reporters driven by IFN-β (pGL3-IFN-β-Luc) or MBT1 (pGL3-MBT1-Luc), and pRL-TK. After 24 h, cells were harvested for luciferase activity analysis. (**C**) IE180 upregulates IFN-I and ISG transcription. Hep2 cells were transfected with EV or IE180 for 24 h. Treatments with dAdT or IFN-α for 8 h served as positive controls for IFN signaling activation. mRNA levels of IFN-α, IFN-β, IRF9, ISG15, and ISG56 were measured by qRT-PCR. (**D**) IE180 enhances STAT1/2 phosphorylation and ISG protein expression. Hep2 cells were transfected with EV or IE180 for 24 h, followed by treatment with PBS, poly(dA:dT), or IFN-α (1000 U/mL) for 12 h. Protein levels of P-STAT1, P-STAT2, IRF9, ISG15, and IE180 were assessed by Western blot (left). GAPDH served as a loading control. The protein levels of P-STAT1, P-STAT2, IRF9, and ISG15 were quantified by densitometry and normalized to the levels of GAPDH (right). (**E**) IE180 promotes STAT1/2 phosphorylation in a time-dependent manner. Hep2 cells were transfected with EV or IE180 for the indicated durations, and P-STAT1, P-STAT2, and IE180 levels were analyzed by Western blot. (**F**) IE180 enhances STAT1/2 phosphorylation in MDCK cells. MDCK cells were transfected with EV or IE180 for 24 h, followed by Western blot analysis of P-STAT1, P-STAT2, and IE180. The protein levels of P-STAT1 and P-STAT2 were quantified by densitometry and normalized to the levels of GAPDH. Data represent mean ± SD from three independent experiments. Statistical analyses were performed using ANOVA or *t*-tests (ns, not significant, *, *p* < 0.05, *** *p* < 0.001, **** *p* < 0.0001).

**Figure 3 microorganisms-13-01397-f003:**
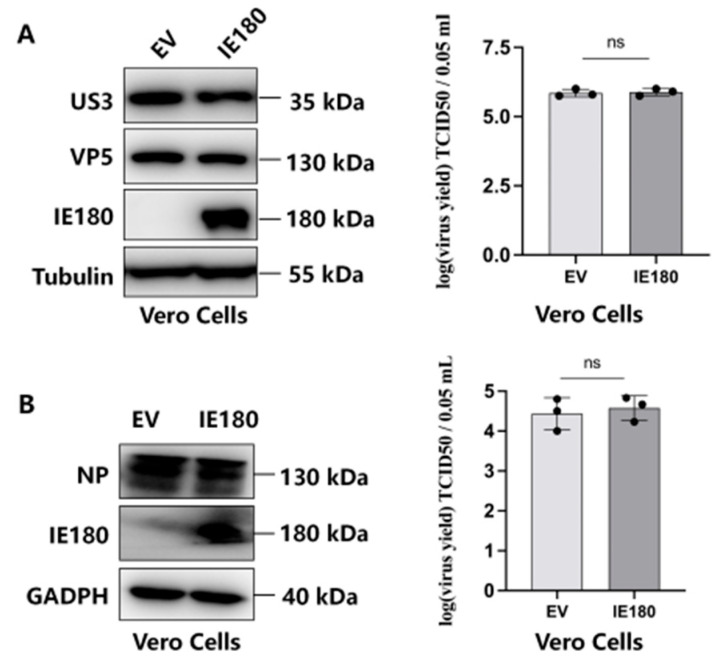
IE180 Loses Its Antiviral Function in Vero Cells. Vero cells were transfected with EV or IE180 for 24 h, followed by PRV (**A**) or H1N1 (PR8) infection (**B**) (MOI = 1). After 24 h, viral protein levels (VP5, US3, and NP) were analyzed by Western blot (left), and viral titers in supernatants were measured using the TCID50 assay (right). Data represent mean ± SD from three independent experiments. Statistical significance was determined using ANOVA (ns, not significant).

**Figure 4 microorganisms-13-01397-f004:**
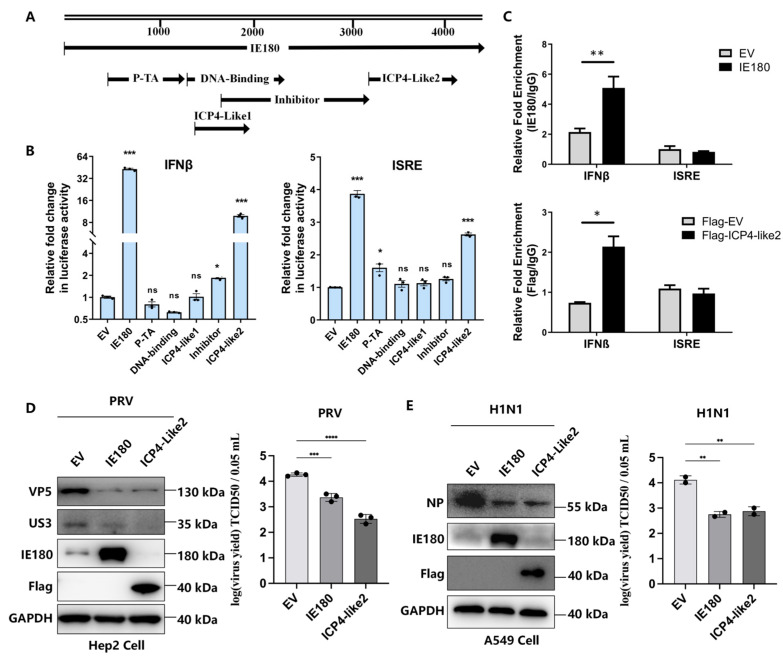
The ICP4-Like2 Domain Mediates IFN-β Activation and Viral Suppression. (**A**) Schematic representation of IE180 functional domains and truncation mutants. (**B**) 293T cells were cotransfected with EV, full-length IE180, or truncation mutants (P-TA, DNA-binding, ICP4-Like1, inhibitor, or ICP4-Like2), along with pGL3-IFNβ-Luc (left) or pGL3-ISRE-Luc (right), plus pRL-TK. Luciferase activity was measured after 24 h. (**C**) IE180 and ICP4-Like2 bind directly to the IFN-β promoter. HEK293T cells were transfected with EV, IE180 (upper panel), or Flag-tagged ICP4-Like2. Chromatin immunoprecipitation (ChIP) was performed 24 h later using anti-IE180 or anti-Flag antibodies to analyze binding to IFN-β or ISRE (from ISG15). (**D**) IE180 and ICP4-Like2 suppress PRV replication. Hep2 cells were transfected with EV, IE180, or ICP4-Like2 for 24 h, followed by PRV infection (MOI = 1). Viral protein levels (VP5, US3) were analyzed by Western blot (left), and viral titers in supernatants were quantified using TCID50 (right). (**E**) IE180 and ICP4-Like2 suppress H1N1 replication. A549 cells were transfected with EV, IE180, or ICP4-Like2 for 24 h, followed by H1N1 (PR8) infection (MOI = 1). Viral NP protein expression was assessed by Western blot (left), and viral titers were measured using TCID50 (right). Data represent mean ± SD from three independent experiments. Statistical significance was determined using ANOVA (ns, not significant, * *p* < 0.05, ** *p* < 0.01, *** *p* < 0.001, **** *p* < 0.0001).

## Data Availability

The original contributions presented in the study are included in the article, further inquiries can be directed to the corresponding author.

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
