# Peer review of "Pseudorabies Virus IE180 Inhibits Virus Replication by Activating the Type I Interferon Pathway"

_microorganisms, 2025, doi:10.3390/microorganisms13061397_

Round 1
Reviewer 1 Report (Previous Reviewer 1)
Comments and Suggestions for Authors
Feiyang Zheng et al. have revised the manuscript and addressed the suggested points. However, there are a few remaining issues that need to be addressed:
Comment 1:
In Figures 2E and 2F, the data indicate that IE180 promotes STAT1/2 phosphorylation in a time-dependent manner in both Hep-2 and MDCK cells. However, IE180 expression is observed more under the EV or (−) condition in MDCK cells than in Hep-2 cells. The authors should mention this in the text. Is the difference due to variations in infection efficiency, viral protein expression, or distinct kinetics of IE180 expression between these cell types? Also, the +/- can be replaced with EV and IE180, like other figures.
Comment 2:
In Figure 3, the label should be corrected from "vero cell" to "Vero cells.
Author Response
Comment 1: In Figures 2E and 2F, the data indicate that IE180 promotes STAT1/2 phosphorylation in a time-dependent manner in both Hep-2 and MDCK cells. However, IE180 expression is observed more under the EV or (−) condition in MDCK cells than in Hep-2 cells. The authors should mention this in the text. Is the difference due to variations in infection efficiency, viral protein expression, or distinct kinetics of IE180 expression between these cell types? Also, the +/- can be replaced with EV and IE180, like other figures.
Response 1: We appreciate the reviewer’s insightful comment regarding the differences in IE180 expression observed in Figures 2E and 2F. In response, we have revised the Results section to acknowledge that IE180 expression under the EV (negative control) condition appears higher in MDCK cells compared to Hep-2 cells. We also discuss that this observation may reflect differences in viral infection efficiency, or distinct kinetics of viral protein expression.
Additionally, as suggested, we have replaced the “+/−” labels in Figures 2E and 2F with “IE180/EV” to maintain consistency with the labeling used in other figures.
Changes have been made in the Results section (lines 222-231) and in the figure panels accordingly.
Comment 2: In Figure 3, the label should be corrected from "vero cell" to "Vero cells..
Response 2: Thank you for pointing this out. We have, accordingly, corrected the labels from “Vero cell” to “Vero cells” in Figure 3.
Reviewer 2 Report (Previous Reviewer 2)
Comments and Suggestions for Authors
I was already positive for publication of this study. The authors now corrected some minor deficiencies as well, therefore I recommend publication in current form
Author Response
Comments: I was already positive for publication of this study. The authors now corrected some minor deficiencies as well, therefore I recommend publication in current form
Response: We sincerely thank the reviewer for the positive evaluation and supportive recommendation. We appreciate your time and feedback throughout the review process.
This manuscript is a resubmission of an earlier submission. The following is a list of the peer review reports and author responses from that submission.
Round 1
Reviewer 1 Report
Comments and Suggestions for Authors
Feiyang Zheng et al. report an interesting finding about the immunomodulatory role of PRV IE180 and its role in viral replication. There are a few things that need to be addressed clearly:
Comment 1: In Figure A, the authors discuss the role of IE180 in regulating viral replication through a time-growth kinetic experiment by detecting viral protein levels of VP5 and US3. However, Figure A lacks VP5 protein data; please include the VP5 data.
Comment 2: The Materials and Methods section lacks required details on using the influenza A virus (H1N1). Authors should include information on the virus strain, source, and propagation in the "Cell Culture and Viruses" subsection (line 76) and infection protocols and titer determination procedures in section 2.6, "Virus Infection and Titer Determination" (line 131). Reporting such details is essential in the interest of reproducibility and scientific integrity.
Comment 3:
In Figure 2D, the authors show that IE180 enhanced STAT1, STAT2 phosphorylation, and IRF9 and ISG15 expression (lines 194 to 196). A quantitative analysis of the Western blot is required to strengthen this data further. The time-course experiment in MDCK cells (Figure 2F) is also missing, which would be essential to establish that IE180 operates in a cell–type–independent fashion.
Comment 4: The plots are not correctly labeled in the Right panels of Figure 4. Each plot must be titled by the corresponding virus name for clarity and to make it reader-friendly.
Comment 5: The authors should keep the text consistent and explain the abbreviation when used for the first time.
Author Response
Comment 1: In Figure A, the authors discuss the role of IE180 in regulating viral replication through a time-growth kinetic experiment by detecting viral protein levels of VP5 and US3. However, Figure A lacks VP5 protein data; please include the VP5 data.
Response 1: In response to the reviewer’s suggestion, we have included the VP5 protein data in Figure1 A to better illustrate the role of IE180 in regulating viral replication.
Comment 2: The Materials and Methods section lacks required details on using the influenza A virus (H1N1). Authors should include information on the virus strain, source, and propagation in the "Cell Culture and Viruses" subsection (line 76) and infection protocols and titer determination procedures in section 2.6, "Virus Infection and Titer Determination" (line 131). Reporting such details is essential in the interest of reproducibility and scientific integrity.
Response 2: Thank you for your insightful comment. In response, we have revised the Materials and Methods section to include detailed information regarding the influenza A virus (H1N1), including the virus strain, source, propagation method, infection protocols, and titer determination procedures. These additions have been made in the "Cell Culture and Viruses" subsection (line 90-91) and section 2.6, "Virus Infection and Titer Determination" (line153-161), as suggested.
Comment 3: In Figure 2D, the authors show that IE180 enhanced STAT1, STAT2 phosphorylation, and IRF9 and ISG15 expression (lines 194 to 196). A quantitative analysis of the Western blot is required to strengthen this data further. The time-course experiment in MDCK cells (Figure 2F) is also missing, which would be essential to establish that IE180 operates in a cell–type–independent fashion.
Response 3: Thank you for your valuable suggestions. In response, we have performed quantitative analyses of the Western blot results shown in Figure 2D (right panels) to better support our findings regarding the enhancement of STAT1 and STAT2 phosphorylation, as well as IRF9 and ISG15 expression by IE180. The densitometric analysis results have been added to the revised figure and described in the corresponding figure legend and main text.
Additionally, we have conducted a time-course experiment in MDCK cells to assess the activation of the JAK-STAT pathway by IE180 over time. The results, now presented in the revised Figure 2F, demonstrate a similar pattern of STAT1/2 phosphorylation and ISG induction as observed in other cell types, supporting the cell-type-independent effect of IE180.
Comment 4: The plots are not correctly labeled in the Right panels of Figure 4. Each plot must be titled by the corresponding virus name for clarity and to make it reader-friendly.
Response 4: We appreciate the reviewer’s comments. We have made the necessary revisions as suggestion. Specifically, we have updated the labels in the text and panels of Figure 4, ensuring that each plot is titled with the corresponding virus name for improved clarity and reader-friendliness.
Comment 5: The authors should keep the text consistent and explain the abbreviation when used for the first time.
Response 5: We appreciate the reviewer’s suggestion. In response, we have carefully reviewed the manuscript to ensure consistency and have provided full definitions for all abbreviations at their first occurrence throughout the text.

Reviewer 2 Report
Comments and Suggestions for Authors
The study provides useful data for the pathogenicity of Pseudorabies virus, an important pathogen of pigs. Overexpression seems to be sufficiently justified, based on both mRNA and protein levels. I suggest publication, but I have some minor comments to be implemented.
The introduction is too generic without providing information to the reader regarding the particular experimentation conducted. I recommend the addition of one paragraph before the scope which should also be divided in a separate paragraph
In Materials and Methods, section 2.4, information regarding quantification methodology is lacking.
Additionally, in the same section, the authors have to add info for primers design. Or, if they were obtained from a previous study, corresponding references.
Finally, in the discussion, there is only one paragraph comparing IE180 with previous studies. Please add more similar references comparing IE180 with previous studies in pigs.
Author Response
Comment 1: The introduction is too generic without providing information to the reader regarding the particular experimentation conducted. I recommend the addition of one paragraph before the scope which should also be divided in a separate paragraph
Response 1: We thank the reviewer for this valuable suggestion. In accordance with the recommendation, we have revised the Introduction section by adding a new paragraph that summarizes the specific experimental approaches conducted in this study (lines 71-77).
Comment 2: In Materials and Methods, section 2.4, information regarding quantification methodology is lacking.
Response 2: We appreciate the reviewer’s insightful suggestion. In response, we have added a detailed description of the quantification methodology used for qPCR analysis in Section 2.4 (lines 113-120) to clarify how relative expression levels were calculated.
Comment 3: Additionally, in the same section, the authors have to add info for primers design. Or, if they were obtained from a previous study, corresponding references.
Response 3: We thank the reviewer for the helpful comment. As suggested, we have added detailed information regarding primer design in Section 2.4 (lines 121-130) .
Comment 4: Finally, in the discussion, there is only one paragraph comparing IE180 with previous studies. Please add more similar references comparing IE180 with previous studies in pigs.
Response 4: We appreciate the reviewer’s suggestion to include more references comparing IE180 with previous studies in pigs. However, we would like to note that current research on IE180, particularly in its natural host (swine), remains limited. This is largely due to the fact that IE180 is an essential gene for PRV replication, and viruses lacking IE180 are replication-defective, making in vivo studies on endogenous IE180 function technically challenging. Given these constraints, only a few studies have investigated IE180’s role in pigs.
In our current discussion, we have incorporated the most relevant and up-to-date findings regarding IE180’s interaction with host cells, including its involvement in immune modulation and stress granule formation. We believe these sufficiently contextualize our findings within the current body of knowledge. Therefore, we respectfully suggest that further expansion may not be necessary at this stage.
